# Enhancing Multimodal Survival Prediction with Pathology Reports in Hyperbolic Space

## Abstract

Cancer survival prediction using computational pathology has emerged as a crucial tool for diagnosis and treatment planning. Current approaches primarily rely on Whole Slide Images (WSIs) and genomic data, but face significant challenges in capturing the logical relationships between visual features and survival outcomes with limited supervision. While pathology reports could potentially serve as a semantic bridge between WSIs and survival time, existing methods overlook the inherent hierarchical relationships between textual descriptions and visual features, where pathology terms represent more abstract concepts and individual terms may correspond to multiple image regions. To address these challenges, we propose HyperSurv, a novel framework that leverages hyperbolic geometry to model the hierarchical relationships between WSIs and pathology reports. Our key insight is that hyperbolic space naturally captures both the entailment structure between generic report concepts and specific visual features, as well as the one-to-many relationships between pathology terms and image regions. HyperSurv enforces these relationships through hyperbolic cones while identifying survival-relevant features via attention pooling. Extensive experiments on four TCGA cancer datasets demonstrate that our approach achieves state-of-the-art survival prediction performance by effectively modeling these multi-modal hierarchical relationships.

## 1 Introduction

Survival prediction plays a crucial role in clinical practice for evaluating mortality risks and treatment outcomes. While **W**hole **S**lide **I**mages (WSIs) provide fine-grained biopsy morphology, genomic data offers complementary molecular profiles essential for comprehensive survival prediction. In addition, the integration of these modalities is capable of revealing unique biomarkers (Chen et al., 2021; Xu & Chen, 2023; Xiong et al., 2024a), enabling more informed clinical decisions.

The primary challenge in this task lies in the *complex logic chain between WSIs and survival outcomes*. Prognosis is typically a complex process of logical deduction incorporating factors such as tumor grading, cancer subtyping, staging, and genetic mutation analysis to estimate the survival time (Bi et al., 2019). Given the gigapixel dimensions of WSIs, WSI-related tasks are often formulated under **M**ultiple **I**nstance **L**earning (MIL) (Lu et al., 2021; Shao et al., 2021; Zhang et al., 2022; Xiong et al., 2023), treating WSIs as bags of instances. Within MIL, the models have to detect discriminative features in an unsupervised manner, which is even harder in this scenario due to the sophisticated logical chain connecting histopathology to survival outcomes.

Pathology reports offer a promising solution by providing structured descriptions of histological features and their clinical significance, serving as a semantic bridge between WSIs and survival outcomes and potentially mitigating the complex logical gaps between WSIs and survival time that exist in end-to-end MIL training. To effectively integrate pathology reports with WSIs and genomic data, we propose **Tri-M**ixture **o**f **M**ultimodal **E**xperts (Tri-MoME), extending MoME (Xiong et al., 2024a) to handle three modalities. Unlike MoME's alternating approach, Tri-MoME processes all modalities concurrently, producing fused representations in a single pass, enhancing efficiency and allowing for a more comprehensive integration of complementary information from all modalities.

Beyond the integration of these three modalities, another significant challenge emerges: *intricate visual-semantic hierarchy between WSIs and pathology reports*. WSIs provide granular features,

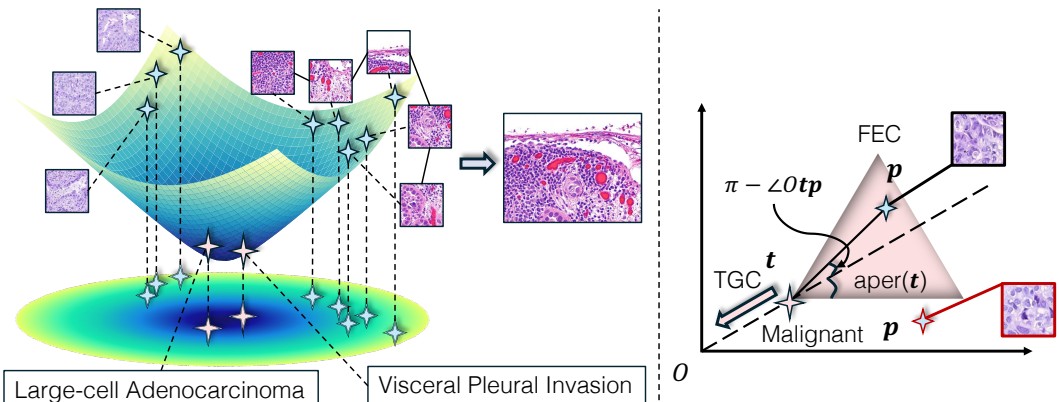

Figure 1: Hyperbolic embedding of lung cancer pathological terminologies and pathological images (left) and geometric representation of our proposed loss function (right).

while pathology reports encapsulate more high-level and generic information that describes the feature morphology. This results in a scenario where the same content is represented by different distributions across the two modalities without hierarchy, posing significant challenges for the model to learn. As illustrated in the left part of Fig. 1, "Large-cell Adenocarcinoma" entails multiple patches within WSIs, and a single terminology like "Visceral Pleural Invasion" requires the integration of multiple images to fully represent the concept. The examples in Fig. 1 also showcase that the text is more generic concepts than image, which has been demonstrated in recent studies (Desai et al., 2023; Ramasinghe et al., 2024). Euclidean space proves inadequate for capturing the visual-semantic hierarchy between image and text, necessitating more suitable representational frameworks (Nickel & Kiela, 2017; 2018; Ganea et al., 2018b; Chen et al., 2022b; Peng et al., 2021; Desai et al., 2023).

To precisely capture the visual-semantic hierarchy, we propose the **HyperSurv** framework, a novel method with Tri-MoME being its backbone and enforcing geometric constraints on text and image features within hyperbolic space. The inherent exponential growth of hyperbolic space naturally accommodates hierarchical structures: images, being more specific, are positioned at the periphery, while terminologies, representing more generic concepts, are situated closer to the origin. Meanwhile, entailment relationships can be effectively expressed using hyperbolic cones (Ganea et al., 2018b), enabling the enforcement of a partial order, "text entails images" (Desai et al., 2023).

As shown on the right side of Fig. 1, we introduce two constraints, **F**eature-pooled **E**ntailment **C**onstraint (FEC) and **T**ext **G**enericity **C**onstraint (TGC). The FEC enforces the entailment by positioning every survival-relevant image within the hyperbolic cone of at least one survival-relevant word. To ensure the accuracy of this constraint on survival-relevant features, we employ an attention pooling module, which calculates survival-relevance scores based on the input features, and selects features with the highest scores. The TGC enforces the generic-specific hierarchy by drawing text vectors closer to the origin while maintaining the original positions of image vectors. Together, our approach significantly enhances the ability of the model to capture the visual-semantic hierarchy.

The contributions of this paper can be summarized as follows:

1. We pioneer the integration of pathology reports, as a visual-semantic bridge, into survival prediction, that could potentially bridge potential logic gaps in end-to-end MIL training.

2. We propose HyperSurv for survival prediction, leveraging hyperbolic geometry to capture the visual-semantic hierarchy across WSIs and pathology reports in terms of loss functions.

3. We validate our approach on four TCGA datasets (BLCA, UCEC, LUAD, and BRCA), demonstrating superior performance in survival prediction across diverse cancer types.

## 2 RELATED WORK

### 2.1 UNIMODAL SURVIVAL PREDICTION

The previous works bifurcates into two categories: *genomic-based* and *WSI-based* approaches.

In WSI-based methods, due to the prohibitive size of WSIs, MIL is employed. This technique involves segmenting WSIs into "bags" of patches, which are then processed using a non-trainable, pre-trained image encoder such as (He et al., 2016), ViT (Dosovitskiy et al., 2021), CTransPath (Wang et al., 2021a), UNI (Chen et al., 2024), CONCH (Lu et al., 2024), and GPFM (Ma et al., 2024). The extracted features from these encoders are subsequently utilized for prediction. There are two predominant strategies for deriving final outcomes in MIL: *instance-level* and *embedding-level* approaches. Instance-level methods (Hou et al., 2016; Campanella et al., 2019; Kanavati & Tsuneki, 2021) involve generating predictions for each instance initially and then aggregating these results. Conversely, embedding-level approaches (Ilse et al., 2018; Lu et al., 2021; Li et al., 2021; Shao et al., 2021; Zhang et al., 2022; Xiong et al., 2023; 2024b) focus on generating a comprehensive bag feature representation before final predictions. Evidence suggests that embedding-level approaches generally outperform instance-level methods (Wang et al., 2018).

In genomic-based approaches, genomic data is typically formatted as $1 \times 1$ tabular representations of gene expressions. Given the straightforward nature of this data structure, various neural network architectures are employed for feature extraction. These include **M**ulti**L**ayer **P**erceptron (MLP), **S**elf-Normalizing **N**eural **N**etworks (SNNs) (Klambauer et al., 2017), and Transformers (Vaswani et al., 2017). These models effectively interpret the genomic data, facilitating subsequent analysis.

### 2.2 MULTIMODAL SURVIVAL PREDICTION

While unimodal methods have shown impressive results in survival prediction, clinical practice often involves multiple data sources, such as WSIs and genomics. Multimodal methods can be categorized into *tensor-based* and *attention-based* methods. Tensor-based methods utilize tensor operations to integrate information across modalities, employing techniques like concatenation (Mobadersany et al., 2018), Kroncecker product (Wang et al., 2021b), and bilinear pooling (Li et al., 2022). These methods generally have fewer parameters (or no parameters) and may exhibit suboptimal performance compared to the attention-based ones. In contrast, the prevailing attention-based methods, which include frameworks such as MCAT (Chen et al., 2021), HMCAT (Li et al., 2023), MOT-CAT (Xu & Chen, 2023), CMTA (Zhou & Chen, 2023), MoME (Xiong et al., 2024a) and SurvPath (Jaume et al., 2024), possess much more parameters. This allows them to more effectively model correlations between modalities, potentially leading to enhanced performance in survival survival.

### 2.3 MULTIMODAL LEARNING IN HYPERBOLIC SPACE

Recent studies (Krioukov et al., 2010; Nickel & Kiela, 2018) have demonstrated that hyperbolic space is particularly suitable for data exhibiting hierarchical structures. Several research has utilized hyperbolic geometry in multimodal learning, showcasing their superiority in capturing the inherent hierarchical structure of image-text data (Desai et al., 2023; Ramasinghe et al., 2024; Ibrahimi et al., 2024; Mandica et al., 2024). However, these studies primarily focus on natural image-text datasets, overlooking the unique characteristics of survival data, which lack ground-truth annotations for paired WSI patches and pathology terminologies. Zhang et al. (2023) conducted multimodal fusion in hyperbolic space for mild cognitive impairment while ignoring the relationship between the hierarchy of the crossed domains. In short, the utilization of hyperbolic geometry to facilitate multimodal survival prediction remains underexplored and merits further investigation.

## 3 PRELIMINARIES

### 3.1 DATA PRE-PROCESSING AND FEATURE EXTRACTION

**WSI.** Under MIL, WSIs are cropped into patches, and then encoded into feature bags with pre-trained vision models (Ilse et al., 2018; Shao et al., 2021; Xiong et al., 2023). The resulting representation is $\boldsymbol{P} \in \mathbb{R}^{n_{\boldsymbol{P}} \times d_{\boldsymbol{P}}}$, where $n_{\boldsymbol{P}}$ is the number of patches and $d_{\boldsymbol{P}}$ is the feature dimension.

**Genomics.** Each dimension of the genomic data represents a gene expression level. Following previous works (Chen et al., 2021; Xu & Chen, 2023; Zhou & Chen, 2023; Xiong et al., 2024a), we group genes into six functional groups that are relevant to cancer progression: Tumor Suppression, Oncogenesis, Protein Kinases, Cellular Differentiation, Transcription, and Cytokines and Growth. To unify the dimensions of variably-sized groups, we project vectors from each group to the same dimension through a **S**elf-normalizing **N**eural **N**etwork (SNN) (Klambauer et al., 2017). The resulting vectors are stacked as a matrix $\boldsymbol{G} \in \mathbb{R}^{n_G \times d_G}$, where $n_{\boldsymbol{G}} = 6$ is the number of groups.

**Pathology Report.** We pre-process pathology reports using Qwen2 (Yang et al., 2024a), one of the **L**arge **L**anguage **M**odels (LLMs), such as GPT4 (Achiam et al., 2023), Llama 3 (Dubey et al., 2024), to extract the survival-relevant texts. The refined text is then embedded using BioClinicalBERT (Alsentzer et al., 2019), yielding a matrix $\boldsymbol{T} \in \mathbb{R}^{n_T \times d_T}$, where $n_{\boldsymbol{T}}$ is the report length.

A final linear transformation is applied to map all features to a uniform dimension $d$. Consequently, the dimensions of the matrices are standardized to $\boldsymbol{P} \in \mathbb{R}^{n_P \times d}, \boldsymbol{G} \in \mathbb{R}^{n_G \times d}$, and $\boldsymbol{T} \in \mathbb{R}^{n_T \times d}$.

## 3.2 Hyperbolic Geometry of Lorentz Model

Hyperbolic geometry is a Riemannian manifold with a constant negative sectional curvature (Nickel & Kiela, 2017; 2018; Ganea et al., 2018a; Peng et al., 2021; Yang et al., 2024b). Among the various mathematical representations of hyperbolic space, the Lorentz manifold, also known as the hyperboloid model, emerged as a prominent one in the machine learning community (Chen et al., 2022b; Sun et al., 2021; Yang et al., 2022) due to its superior stability in numerical optimization processes.

**Definition 1** (Lorentzian Inner Product). *The inner product $\langle \boldsymbol{x}, \boldsymbol{y} \rangle_{\mathcal{L}}$ for $\boldsymbol{x}, \boldsymbol{y} \in \mathbb{R}^{d+1}$ is given by,*

$$\langle \boldsymbol{x}, \boldsymbol{y} \rangle_{\mathcal{L}} = -x_0 y_0 + \sum_{i=1}^{d} x_i y_i. \tag{1}$$

**Definition 2** (Lorentz Manifold). *A $d$-dimensional Lorentz manifold $\mathbb{L}^d$ with a negative curvature of $-1$ can be defined as the Riemannian manifold $\left( \mathbb{H}^d, g_\ell \right)$, where,*

$$\mathbb{H}^d = \left\{ \boldsymbol{x} \in \mathbb{R}^{d+1} : \langle \boldsymbol{x}, \boldsymbol{x} \rangle_{\mathcal{L}} = -1, x_0 > 0 \right\}, \quad g_\ell = \mathrm{diag}([-1, 1, \ldots, 1]). \tag{2}$$

**Definition 3** (Lorenzian Distance). *For two points $\boldsymbol{x}, \boldsymbol{y} \in \mathbb{L}^d$, the Lorenzian distance is defined as,*

$$d_{\mathcal{L}}(\boldsymbol{x}, \boldsymbol{y}) = arcosh(-\langle \boldsymbol{x}, \boldsymbol{y} \rangle_{\mathcal{L}}). \tag{3}$$

For computational efficiency, a widely adopted approach in mapping vectors between Euclidean space and hyperbolic space (Nickel & Kiela, 2017; 2018) is to designate the origin of the Lorentz manifold, $\boldsymbol{o}^{\mathbb{L}} = (1, 0, 0, ..., 0) \in \mathbb{L}^d$, as the reference point, facilitating simplified expressions for the exponential maps ($\exp_{\boldsymbol{o}} : \mathbb{R}^{d+1} \to \mathbb{L}^d$) and logarithmic maps ($\log_{\boldsymbol{o}} : \mathbb{L}^d \to \mathbb{R}^{d+1}$) as follows,

$$\exp_{\boldsymbol{o}}(\boldsymbol{v}) = \exp_{\boldsymbol{o}}([0, \boldsymbol{v}]) = \left( \cosh\left( \|\boldsymbol{v}\|_2 \right), \sinh\left( \|\boldsymbol{v}\|_2 \right) \frac{\boldsymbol{v}}{\|\boldsymbol{v}\|_2} \right), \tag{4}$$

$$\log_{\boldsymbol{o}}(\boldsymbol{x}^{\mathbb{L}}) = d_{\mathcal{L}}(\boldsymbol{o}^{\mathbb{L}}, \boldsymbol{x}^{\mathbb{L}}) \frac{\boldsymbol{x}^{\mathbb{L}} + \langle \boldsymbol{o}^{\mathbb{L}}, \boldsymbol{x}^{\mathbb{L}} \rangle_{\mathcal{L}} \boldsymbol{o}^{\mathbb{L}}}{\|\boldsymbol{x}^{\mathbb{L}} + \langle \boldsymbol{o}^{\mathbb{L}}, \boldsymbol{x}^{\mathbb{L}} \rangle_{\mathcal{L}} \boldsymbol{o}^{\mathbb{L}}\|_{\mathcal{L}}}, \tag{5}$$

where $[,]$ denotes concatenation. For clarity, $\cdot^{\mathbb{L}}$ denotes hyperbolic space features, while Euclidean embeddings are without superscripts. It is noteworthy that the first dimension of the vectors in the tangent space of the origin is 0 (Sun et al., 2021). Therefore, they will be dropped after $\log_{\boldsymbol{o}}(\cdot^{\mathbb{L}})$.

## 3.3 Survival Prediction Formulation

We estimate the survival probability at time $t$, rather than predicting the exact death time (Chen et al., 2021). This approach accommodates right-censored clinical data as the incomplete patient follow-ups are inevitable. Let $T$ be the time until death, the hazard function $h(t|\boldsymbol{P}, \boldsymbol{G}, \boldsymbol{T})$, describing the instantaneous death rate at time $t$ conditional on survival up to that time, can be expressed as,

$$h(t|\boldsymbol{P}, \boldsymbol{G}, \boldsymbol{T}) = \lim_{\Delta t \to 0} \frac{\Pr(t \leq T < t + \Delta t | T \geq t, (\boldsymbol{P}, \boldsymbol{G}, \boldsymbol{T}))}{\Delta t}, \tag{6}$$

where $\Pr(\cdot)$ denotes probability. The survival function, $S(t|\boldsymbol{P},\boldsymbol{G},\boldsymbol{T})$, indicating the probability of surviving beyond time $t$, is derived from the hazard function, which can be expressed as,

$$S(t|\boldsymbol{P},\boldsymbol{G},\boldsymbol{T}) = \exp\left(-\int_0^t h(u|\boldsymbol{P},\boldsymbol{G},\boldsymbol{T})du\right). \tag{7}$$

This formulation is the exponential form of the survival function. To apply this framework, we employ the **N**egative **L**og-**L**ikelihood (NLL) loss (Chen et al., 2021). Let $\delta$ denote the event indicator ($\delta = 0$ if the event occurred, $\delta = 1$ if the data is right-censored), and $\alpha$ denote a weighting factor balancing censored and uncensored loss components. Our survival loss function $\mathcal{L}_s$ is given by,

$$\mathcal{L}_s = (1-\delta)\big(\log S(t|\boldsymbol{P},\boldsymbol{G},\boldsymbol{T}) + \log h(t|\boldsymbol{P},\boldsymbol{G},\boldsymbol{T})\big) + \delta(1-\alpha)\log S(t+1|\boldsymbol{P},\boldsymbol{G},\boldsymbol{T}). \tag{8}$$

## 4 METHODOLOGY

HyperSurv consists of four layers of Tri-MoME and two constraints in hyperbolic space, namely FEC and TGC. Given the absence of a tri-modal backbone for survival prediction, we first adapt MoME to Tri-MoME, then introduce these two hyperbolic constraints.

### 4.1 TRI-MOME

**Tri-MoME Architecture.** While MoME (Xiong et al., 2024a) alternately encodes and fuses each modality, this method becomes inefficient as the number of modalities increases, particularly with the resource-intensive Transformer backbone. To this end, we propose Tri-MoME to simultaneously encode and fuse all three modalities. We encode all modalities four times, resulting in four Tri-MoME layers. Let $i$ denote the number of encoding iterations; the Tri-MoME layer is given by,

$$\boldsymbol{P}^{(i+1)},\boldsymbol{G}^{(i+1)},\boldsymbol{T}^{(i+1)} = \text{Tri-MoME}(\boldsymbol{P}^{(i)},\boldsymbol{G}^{(i)},\boldsymbol{T}^{(i)}). \tag{9}$$

Same as MoME (Xue & Marculescu, 2023; Xiong et al., 2024a), only one expert is activated from the following experts in each layer, unlike the traditional MoEs (Masoudnia & Ebrahimpour, 2014). Readers interested in more details are referred to the original paper of MoME (Xiong et al., 2024a).

**Gating Network.** The gating network has been revised to accept inputs from three modalities and output the logits for the selection of experts. Mathematically, the logits are obtained through,

$$\text{logits} = \boldsymbol{W}\cdot\big(g(\boldsymbol{P}^{(i)})+g(\boldsymbol{G}^{(i)})+g(\boldsymbol{T}^{(i)})\big); g(\boldsymbol{P}^{(i)}) = \text{mean}\big(\text{GELU}(\text{RMS}_{\boldsymbol{P}}(\boldsymbol{W}_{\boldsymbol{P}}\boldsymbol{P}^{(i)}))\big), \tag{10}$$

where $\boldsymbol{W}$ and $\boldsymbol{W}_{\boldsymbol{P}}$ are two learnable matrices, $g(\cdot)$ is the aggregation function which takes a matrix as input and outputs a vector representing that matrix, $g(\boldsymbol{T})$ and $g(\boldsymbol{G})$ are defined analogously to $g(\boldsymbol{P})$, GELU$(\cdot)$ is the **G**aussian **E**rror **L**inear **U**nits (GELUs) (Hendrycks & Gimpel, 2016), and RMS.$(\cdot)$ is the **R**oot **M**ean **S**quare (RMS) layer normalization layers (Zhang & Sennrich, 2019).

**TransFusion.** Given the input $(\boldsymbol{P}^{(i)},\boldsymbol{G}^{(i)},\boldsymbol{T}^{(i)})$, the pathology image component $\boldsymbol{P}^{(i+1)}$ of the **Trans**Fusion (TF) expert output is defined as,

$$\boldsymbol{P}^{(i+1)} = \text{TF}_{\boldsymbol{P}}(\boldsymbol{P}^{(i)},\boldsymbol{G}^{(i)},\boldsymbol{T}^{(i)}) = \text{SA}([\boldsymbol{P}^{(i)},\boldsymbol{G}^{(i)},\boldsymbol{T}^{(i)}])[:n_{\boldsymbol{P}},:], \tag{11}$$

where SA$(\cdot)$ denotes the **S**elf-**A**ttention (Vaswani et al., 2017). The genomic component $\boldsymbol{G}^{(i+1)}$ and the pathology report component $\boldsymbol{T}^{(i+1)}$ are derived similarly, with appropriate slicing operations.

**SNNFusion.** The **SNN**Fusion (SNF) expert utilizes three SNNs, denoted as S.$(\cdot)$, and the pathology image component $\boldsymbol{P}^{(i+1)}$ of the SNF expert output can be expressed as as,

$$\begin{aligned}\boldsymbol{P}^{(i+1)} &= \text{SNF}_{\boldsymbol{P}}(\boldsymbol{P}^{(i)},\boldsymbol{G}^{(i)},\boldsymbol{T}^{(i)}),\\ &= \text{S}_{\boldsymbol{P}}(\text{RMS}_{\boldsymbol{P}}(\boldsymbol{P}^{(i)})) + \text{mean}(\text{S}_{\boldsymbol{G}}(\text{RMS}_{\boldsymbol{G}}(\boldsymbol{G}^{(i)}))) + \text{mean}(\text{S}_{\boldsymbol{T}}(\text{RMS}_{\boldsymbol{T}}(\boldsymbol{T}^{(i)}))).\end{aligned} \tag{12}$$

**SkipFusion.** The **S**kip**F**usion (SF) expert serves as a pass-through layer, activated when the gating network determines that the current embedding is adequate. The SF expert is defined as,

$$(\boldsymbol{P}^{(i+1)},\boldsymbol{G}^{(i+1)},\boldsymbol{T}^{(i+1)}) = \text{SF}(\boldsymbol{P}^{(i)},\boldsymbol{G}^{(i)},\boldsymbol{T}^{(i)}) = (\boldsymbol{P}^{(i)},\boldsymbol{G}^{(i)},\boldsymbol{T}^{(i)}). \tag{13}$$

For clarity and consistency throughout the remainder of this paper, we will denote matrices for WSI, genomics, and texts after encoded by Tri-MoMEs as $\boldsymbol{P}$, $\boldsymbol{G}$, and $\boldsymbol{T}$, respectively.

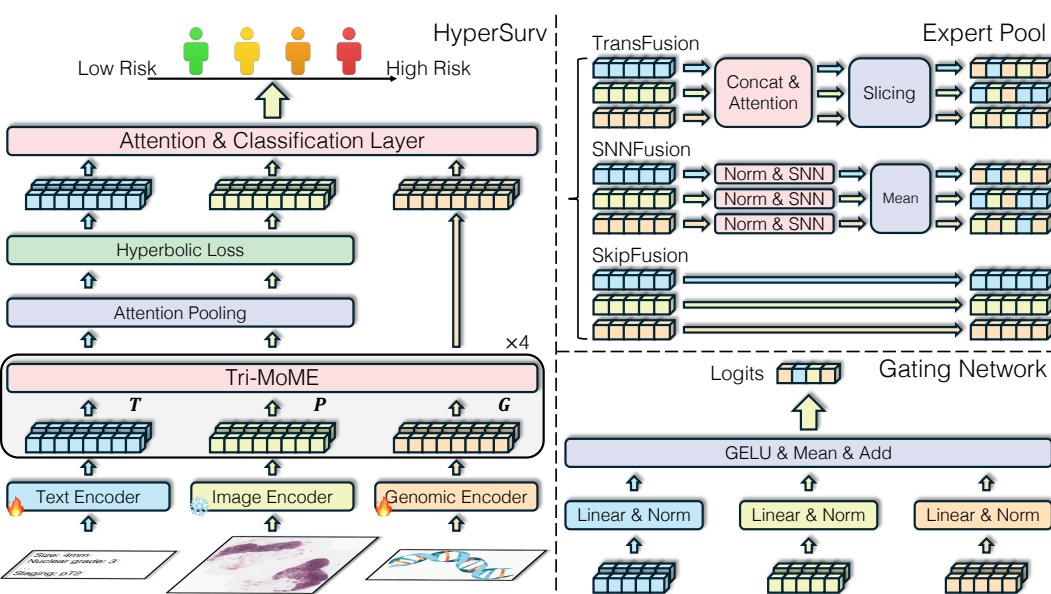

Figure 2: Illustration of HyperSurv (left) and its constituent Tri-MoME layer (right). The Tri-MoME layer comprises two parts: the gating network (bottom right) and the expert pool (top right).

## 4.2 HYPERBOLIC CONSTRAINTS

While the Tri-MoME effectively processes three modalities in Euclidean space, it faces challenges in capturing the visual-semantic hierarchy between pathology reports and WSIs, as illustrated in Fig. 1, due to the inherent deficiencies of Euclidean space. Specifically, we have identified two key characteristics that previous works often ignore: 1) one terminology can entail multiple pathology images, and 2) text usually conveys more generic information than image. To address this challenge, we propose FEC and TGC in Lorentz space to capture this hierarchy, as it provides an effective framework for representing this hierarchy, and we will detail these two constraints in this section.

### 4.2.1 FEATURE-POOLED ENTAILMENT CONSTRAINT

In hyperbolic space, if one concept entails another concept, the vector representing the former is positioned within the hyperbolic cone defined by the latter (Ganea et al., 2018b; Desai et al., 2023). Following this rationale, in our case, if a pathological word entails a specific image feature, the image vector should be situated within the hyperbolic cone defined by the pathological word vector.

Unlike previous works (Desai et al., 2023; Ramasinghe et al., 2024) utilizing ground truth image-text pair, pathology terms and WSI patches exhibit a complex, non-one-to-one association lacking ground truth due to extraneous elements and labor-intensive annotation. To tackle this unique challenge, we propose a feature-pooled entailment loss function. Firstly, we apply attention pooling modules to both text and image features, aiming to identify the survival-relevant features from each modality. By focusing on these survival-relevant features, we ensure that extraneous ones are not involved in this loss and the impact of irrelevant features is effectively eliminated. Mathematically, we express the pooled text matrix, representing the survival-relevant text matrix $\boldsymbol{T}^{\mathbb{L}}_{pooled}$ as follows,

$$\boldsymbol{T}^{\mathbb{L}}_{pooled} = \exp_{\boldsymbol{o}}(\boldsymbol{T}[\boldsymbol{i}, :]), \quad \boldsymbol{i} = \mathrm{argsort}(\boldsymbol{T}\boldsymbol{A_T})[: k_t] \in \mathbb{R}^{k_t}, \quad (14)$$

where $\mathrm{argsort}(\cdot)[: k_t]$ returns the indices of the top-$k_t$ highest values, $k_t$ is the number of output features for text modality after pooling, and $\boldsymbol{A_T} \in \mathbb{R}^{d \times 1}$ is a learnable matrix that assigns attention scores to each instance. $\boldsymbol{P}^{\mathbb{L}}_{pooled}$ is obtained through the same procedure and $k_p$ is defined similarly.

Subsequently, we evaluate whether each survival-relevant image falls within the hyperbolic cone defined by any text vector. When this relationship is not satisfied, we impose our loss function, which is derived from the angular discrepancy between the vectors. More precisely, for each survival-relevant image row vector $\boldsymbol{p}^{\mathbb{L}} \in \mathbb{L}^d$ from $\boldsymbol{P}^{\mathbb{L}}_{pooled}$, we calculate the angular difference between $\boldsymbol{p}^{\mathbb{L}}$

and every row of $T_{pooled}^{\mathbb{L}}$, retrieving the text row vector $t^{\mathbb{L}}$ with the smallest difference. Our approach then compares two angular measurements: the aperture angle defined by $t^{\mathbb{L}}$ and the exterior angle between $t^{\mathbb{L}}$ and $p^{\mathbb{L}}$. This comparison serves as the foundation for our loss, which is given by,

$$\ell_e(t^{\mathbb{L}}, p^{\mathbb{L}}) = \max\left((\pi - \text{angle}(t^{\mathbb{L}}, p^{\mathbb{L}})) - \text{aper}(t^{\mathbb{L}}), 0\right), \ \text{aper}(t^{\mathbb{L}}) = \arcsin\left(\frac{2\beta}{\|t^{\mathbb{L}}\|}\right),$$

$$(\pi - \text{angle}(t^{\mathbb{L}}, p^{\mathbb{L}})) = \arccos\left(\frac{p_0 + t_0\langle t^{\mathbb{L}}, p^{\mathbb{L}}\rangle_{\mathcal{L}}}{\|t^{\mathbb{L}}\|\sqrt{(\langle t^{\mathbb{L}}, p^{\mathbb{L}}\rangle_{\mathcal{L}})^2 - 1}}\right), \tag{15}$$

where $t_0$ and $p_0$ are the 0-th dimension of $t^{\mathbb{L}}$ and $p^{\mathbb{L}}$, respectively, $\beta = 0.1$ sets the boundary condition near the origin. The first term is the exterior angle of $\angle Otp$, representing the angle between $t^{\mathbb{L}}$ and $p^{\mathbb{L}}$, and the second term is the angle of the aperture of the text cone. If the exterior angle is smaller than the angle of the aperture, the image falls within the cone; therefore, we will not punish the model. Otherwise, a loss is calculated through the difference between these two angles. Building upon Eq. (15), we compute the loss for each row vector in $P_{pooled}^{\mathbb{L}}$. The overall FEC loss function $\mathcal{L}_e$ is then derived as the mean of these individual angular losses as follows,

$$\mathcal{L}_e(T_{pooled}^{\mathbb{L}}, P_{pooled}^{\mathbb{L}}) = \frac{1}{k_p} \sum_{i=1}^{k_p} \ell_e\left(\underset{t^{\mathbb{L}} \in T_{pooled}^{\mathbb{L}}}{\arg\min} (\pi - \text{angle}(t^{\mathbb{L}}, P_{pooled}^{\mathbb{L}}[i,:])), P_{pooled}^{\mathbb{L}}[i,:]\right). \tag{16}$$

Given the heavy computational costs required for mapping vectors into hyperbolic space, we only apply this geometric constraint to the final embeddings of the text and images (Desai et al., 2023).

### 4.2.2 Text Genericity Constraint

Pathology reports often encapsulate more generic information than WSIs. This is because a pathology report typically provides an overview of relevant information in WSIs. Previous research has demonstrated that individual images contain more localized and specific information but overall less semantic content compared to individual words (Desai et al., 2023; Ramasinghe et al., 2024). These studies also indicated the efficacy of hyperbolic embeddings in visual-language-related tasks, highlighting their ability when representing complex hierarchies.

To this end, we propose leveraging the Lorentz model to effectively capture this relationship between pathology reports and WSIs, as in hyperbolic space, vectors representing generic concepts are positioned closer to the origin, while more specific entities are situated farther away. This property can naturally accommodate the hierarchical structures between pathology reports and WSIs.

Specifically, we first project the encoded representations from Euclidean space onto hyperbolic space. Within the hyperbolic space, we position the text representations closer to the origin while maintaining the original positions of the image embeddings. Formally, for every text representation in a pathology report $t \in \mathbb{R}^d$ in Euclidean space, we project it to $t^{\mathbb{L}} \in \mathbb{L}_d$ in the Lorentz model using the projection function defined in Eq. (4). Subsequently, based on the distance metric in Eq. (3), we apply the loss function $\mathcal{L}_t$ to every hyperbolic embedding of text representation, which is given by,

$$\mathcal{L}_t = \frac{1}{n_T} \sum_{t \in T} d_{\mathcal{L}}(t^{\mathbb{L}}, o^{\mathbb{L}}) = \frac{1}{n_T} \sum_{t \in T} \text{arcosh}(-\langle \exp_o(t), o^{\mathbb{L}}\rangle_{\mathcal{L}}). \tag{17}$$

We constrain solely pathology report embeddings for two key factors. Firstly, while images predominantly contain specific information, they occasionally can represent generic concepts, rendering it inappropriate to adjust their position relative to the origin. Secondly, imposing distance constraints on images would complicate the model, as controlling the image-text distances is challenging and could potentially obscure the core concept. By focusing on text embeddings, we maintain a clear representation of the hierarchy without introducing unnecessary complexity.

Combining all loss functions from Eqs. (8), (16) and (17), we can obtain our final loss function as,

$$\mathcal{L} = \mathcal{L}_s + \gamma\mathcal{L}_e + \lambda\mathcal{L}_t, \tag{18}$$

where $\gamma$ and $\lambda$ are coefficients which balance these loss functions. Specifically, $\gamma$ adjusts the impact of the FEC loss, while $\lambda$ controls the influence of the TGC loss. These coefficients allow us to fine-tune the trade-off between different aspects of the performance of our model.

Table 1: Comparative C-index performance across multiple methods on four TCGA datasets. The highest C-index scores per dataset are underlined in red, with the second-highest scores italicized in blue. "Mod." indicates modalities utilized: "G" for genomic data, "P" for WSIs, and "T" for pathology reports. The rightmost column presents the averaged performance across all datasets.

| Methods | Mod. | Datasets | | | | Overall |
| | | BLCA | UCEC | LUAD | BRCA | |
|---|---|---|---|---|---|---|
| SNN | G | 0.618±0.022 | 0.679±0.040 | 0.611±0.047 | 0.624±0.060 | 0.633 |
| SNNTrans | G | 0.659±0.032 | 0.656±0.038 | 0.638±0.022 | 0.647±0.063 | 0.650 |
| AttnMIL | P | 0.599±0.048 | 0.658±0.036 | 0.620±0.061 | 0.609±0.065 | 0.622 |
| CLAM-S | P | 0.559±0.034 | 0.644±0.061 | 0.594±0.063 | 0.573±0.044 | 0.593 |
| CLAM-M | P | 0.565±0.027 | 0.609±0.082 | 0.582±0.072 | 0.578±0.032 | 0.584 |
| TransMIL | P | 0.575±0.034 | 0.655±0.046 | 0.642±0.046 | 0.666±0.029 | 0.635 |
| DTFDMIL | P | 0.546±0.021 | 0.656±0.045 | 0.585±0.066 | 0.609±0.059 | 0.599 |
| MCAT | G+P | 0.672±0.032 | 0.649±0.043 | 0.659±0.027 | 0.659±0.031 | 0.660 |
| GPDBN | G+P | 0.636±0.014 | 0.682±0.050 | 0.632±0.059 | 0.660±0.040 | 0.653 |
| Porpoise | G+P | 0.636±0.024 | 0.692±0.048 | 0.647±0.031 | 0.621±0.054 | 0.649 |
| HFBSurv | G+P | 0.640±0.040 | 0.699±0.025 | 0.639±0.027 | 0.653±0.032 | 0.658 |
| MOTCAT | G+P | 0.682±0.023 | 0.671±0.053 | 0.673±0.040 | 0.671±0.021 | 0.674 |
| CMTA | G+P | 0.672±0.038 | 0.691±0.066 | 0.676±0.037 | 0.659±0.013 | 0.675 |
| MoME | G+P | *0.686*±0.041 | *0.706*±0.038 | *0.691*±0.040 | 0.656±0.047 | 0.685 |
| CGM | G+P | *0.686*±0.025 | 0.703±0.048 | 0.696±0.023 | *0.684*±0.034 | *0.691* |
| Ours | G+P+T | 0.701±0.041 | 0.757±0.040 | 0.688±0.046 | 0.691±0.042 | 0.709 |

## 5 EXPERIMENTS AND RESULTS

### 5.1 DATASETS

We use the publicly available data from **T**he **C**ancer **G**enome **A**tlas (TCGA) project[1], which provides WSIs, genomic data, pathology reports, and survival time. Our study utilizes four datasets from this project: 373 samples of **B**ladder **U**rothelial **CA**rcinoma (TCGA-BLCA), 480 samples of **U**terine **C**orpus **E**ndometrial **C**arcinoma (TCGA-UCEC), 453 samples of **LU**ng **AD**enocarcinoma (TCGA-LUAD), and 956 samples from **BR**east Invasive **CA**rcinoma (TCGA-BRCA). Pathology reports are pre-processed (Kefeli & Tatonetti, 2024) and cleaned and refined using Qwen2 (Yang et al., 2024a). However, not all samples include pathology reports, and the availability is 78.3% for BLCA, 88.9% for UCEC, 83.1% for LUAD, and 87.6% for BRCA.

### 5.2 IMPLEMENTATION DETAILS

#### 5.2.1 TRAINING SETTINGS

We evaluate our method against a range of existing approaches, including both unimodal and multi-modal ones. The comparative methods include: SNN (Klambauer et al., 2017), SNNTrans (Klambauer et al., 2017; Vaswani et al., 2017), AttnMIL (Ilse et al., 2018), CLAM (Lu et al., 2021), TransMIL (Shao et al., 2021), DTFD-MIL (Zhang et al., 2022), MCAT (Chen et al., 2021), GPDBN (Wang et al., 2021b), Porpoise (Chen et al., 2022a), HFBSurv (Li et al., 2022), MOTCAT (Xu & Chen, 2023), CMTA (Zhou & Chen, 2023), MoME (Xiong et al., 2024a), and CGM (Zhou et al., 2024). We employ the **C**oncordance index (C-index), which holistically assesses the discriminative power and predictive accuracy of a model. Five-fold cross-validation is used to evaluate the performance of models. Each model is trained for 20 epochs, and the performance on each validation fold is recorded. We report the means and standard deviations of the C-index for each method.

#### 5.2.2 HYPER-PARAMETERS

Our experimental configuration employs the Adam optimizer (Kingma & Ba, 2015) with a learning rate of $2\times10^{-4}$ and weight decay of $1\times10^{-5}$, following (Xu & Chen, 2023). WSIs are segmented

---

[1]https://www.cancer.gov/ccg/research/genome-sequencing/tcga

Table 2: Ablation study results comparing C-index performance across different model configurations and datasets. The first column indicates which components are removed from the full model. $\mathcal{L}_t$ and $\mathcal{L}_e$ represent specific loss components in the model.

| Variants | Mod. | Datasets | | | | Overall |
|---|---|---|---|---|---|---|
| | | BLCA | UCEC | LUAD | BRCA | |
| Ours | G+P+T | 0.701±0.041 | 0.757±0.040 | 0.688±0.046 | 0.691±0.042 | 0.709 |
| $-\mathcal{L}_t$ | G+P+T | 0.687±0.024 | 0.743±0.057 | 0.685±0.031 | 0.687±0.024 | 0.701 |
| $-\mathcal{L}_e$ | G+P+T | 0.686±0.026 | 0.747±0.049 | 0.670±0.028 | 0.661±0.015 | 0.691 |
| $-\mathcal{L}_e - \mathcal{L}_t$ | G+P+T | 0.693±0.039 | 0.739±0.045 | 0.685±0.032 | 0.675±0.052 | 0.698 |
| $-\mathcal{L}_e - \mathcal{L}_t$ | G+P | 0.674±0.054 | 0.738±0.030 | 0.681±0.032 | 0.663±0.054 | 0.689 |
| $-\mathcal{L}_e - \mathcal{L}_t$ | G+T | 0.678±0.046 | 0.701±0.078 | 0.678±0.031 | 0.659±0.062 | 0.679 |

Table 3: Comparative C-index results across different model configurations on four TCGA datasets. Results are presented for various combinations of $k_t$ and $k_p$ values, respectively.

| Variants | Datasets | | | | Overall |
|---|---|---|---|---|---|
| | BLCA | UCEC | LUAD | BRCA | |
| $k_t = 8, k_p = 16$ | 0.670±0.033 | 0.717±0.043 | 0.672±0.038 | 0.685±0.029 | 0.686 |
| $k_t = 8, k_p = 32$ | 0.701±0.041 | 0.757±0.040 | 0.688±0.046 | 0.691±0.042 | 0.709 |
| $k_t = 8, k_p = 64$ | 0.692±0.018 | 0.736±0.036 | 0.681±0.057 | 0.673±0.042 | 0.696 |
| $k_t = 16, k_p = 16$ | 0.687±0.023 | 0.718±0.039 | 0.663±0.035 | 0.687±0.024 | 0.689 |
| $k_t = 16, k_p = 32$ | 0.680±0.037 | 0.727±0.027 | 0.693±0.026 | 0.675±0.014 | 0.694 |
| $k_t = 16, k_p = 64$ | 0.693±0.012 | 0.745±0.076 | 0.681±0.034 | 0.680±0.031 | 0.700 |

into patches of 224×224 pixels at 20× magnification, with features extracted via CTransPath (Wang et al., 2021a). Input dimensions for each modality are set to $d_p = d_t = 768$ and $d_g = 512$, with a hidden dimension of $d = 512$. We employ the micro-batch technique (Xu & Chen, 2023) with a batch size of 4,096 and set the number of encoding iterations to $m = 4$. Loss function coefficients are set to $\alpha = 0$ for censored/uncensored balance, and $\gamma = \lambda = 0.5$ for the final loss. The number of attention pooling output features are set to $k_t = 8$ for pathology reports and $k_p = 32$ for WSIs.

## 5.3 COMPARISON RESULTS

The results of all comparison methods are presented in Table 1. Our method consistently outperforms existing approaches, demonstrating substantial improvements on BLCA, UCEC, and BRCA datasets with C-index increases of 1.5%, 5.1%, and 0.7% respectively, compared to the previous best performing methods. Our method maintains competitive performance, even though it is slightly worse (0.8%) than CGM on the LUAD dataset. Notably, our approach achieves the highest overall performance with a C-index of 0.709, indicating a 1.8% improvement over the second best method, CGM (0.691). This consistent superior performance across diverse cancer types suggests the robustness of our approach. Among multimodal methods, recent approaches like MoME and CGM already show strong performance. However, the incorporation of pathology report data provides a significant advantage, suggesting that pathology reports indeed contribute meaningful predictive power to the model. Overall, the comparison results demonstrate the effectiveness of incorporating pathology reports as the third modality and the efficacy of our proposed method.

## 5.4 ABLATION STUDIES ON LOSS FUNCTIONS

We conduct experiments to validate the effectiveness of each component. The results are presented in Table 2. Our full model (G+P+T) achieves the best overall performance (0.709). Removing the TGC loss $\mathcal{L}_t$ slightly decreases performance (0.701), while ablating the FEC loss $\mathcal{L}_e$ leads to a more significant drop (0.691). This suggests that both losses contribute to the model's effectiveness, and $\mathcal{L}_e$ plays a particularly crucial role. Interestingly, removing both $\mathcal{L}_e$ and $\mathcal{L}_t$ while retaining all modalities (0.698), which is a model purely in Euclidean space, outperforms the model with only $\mathcal{L}_e$ removed, indicating a complex interaction between these loss components, which might need further

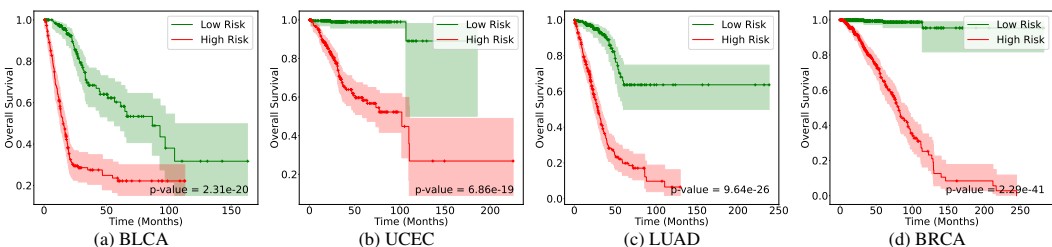

Figure 3: Kaplan-Meier curves of our HyperSurv on the four cancer datasets.

investigation in the future work. The variant without $\mathcal{L}_e$, $\mathcal{L}_t$, and the text modality (T) shows an even worse overall performance (0.689), further underscoring the importance of pathology report as a new modality. Moreover, substituting WSIs with pathology reports, which are considered summaries of WSIs, results in further performance degradation (0.679). This demonstrates the irreplaceable role of WSIs in survival prediction, attributable to their provision of fine-grained, localized features. These results demonstrate the effectiveness of our full model, the significant contributions of both loss components, and the pathology report modality.

### 5.5 SENSITIVITY ANALYSIS ON NUMBER OF POOLED FEATURES

We conduct sensitivity analyses to assess the impact of varying the number of attention pooling features for pathology reports ($k_t$) and WSIs ($k_p$) on model performance. Table 3 shows C-index results for different $k_t$ and $k_p$ values across four TCGA datasets. The configuration $k_t = 8, k_p = 32$ achieved the best overall performance (C-index: 0.709) and outperformed others in three out of four datasets (BLCA, UCEC, and BRCA). This suggests an optimal balance between features extracted from pathology reports and WSIs. Increasing $k_p$ to 64 while maintaining $k_t = 8$ led to a slight performance decrease (0.696), indicating that excessive WSI features may introduce noise. Similarly, increasing $k_t$ to 16 did not improve overall performance, regardless of $k_p$, suggesting that 8 features from pathology reports sufficiently capture survival-relevant information. Notably, the best-performing configuration for LUAD ($k_t = 16, k_p = 32$) differed from the overall best, highlighting potential cancer-specific tuning needs. However, the marginal difference (0.693 vs. 0.688) supports the robustness of the $k_t = 8, k_p = 32$ configuration across cancer types. These results underscore the importance of balancing attention pooling features from different modalities.

### 5.6 KAPLAN–MEIER ANALYSIS

To further validate the differentiability of the model, we conduct a Kaplan–Meier analysis. Patients are stratified into low-risk and high-risk groups based on whether their risk values exceeded the mean risk value of the entire cohort. We then visualize the survival events for all patients in Fig. 3, and perform a log-rank test to assess the statistical significance of the difference between the low-risk and high-risk cohorts. Conventionally, a p-value less than 0.05 is considered statistically significant. As evident from Fig. 3, our method successfully stratified patients into two groups with high statistical significance, demonstrating the robust differentiability of our method.

## 6 CONCLUSION

In this paper, we introduced a novel approach to survival prediction in pathology by incorporating pathology reports and enforcing geometric constraints for pathology reports and WSIs in hyperbolic space. The integration mitigates the complex logical gaps inherent in end-to-end model training from WSIs to survival outcomes. To accommodate the inherent hierarchy between pathology reports and WSIs, we leveraged hyperbolic space, specifically the Lorentz manifold, to overcome the limitations of Euclidean space. Our method enforces the partial order relationship as the entailment between pathology reports and WSIs, and positions text representations closer to the origin to represent more generic concepts. Extensive experiments across multiple datasets demonstrated that our proposed method consistently outperforms previous approaches. These results provide strong evidence for the effectiveness of textual modality and validate the efficacy of our hyperbolic geometric framework.

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
