# OpenReview forum: "Enhancing Multimodal Survival Prediction with Pathology Reports in Hyperbolic Space"
_ICLR.cc/2025/Conference — ICLR 2025 Conference Withdrawn Submission_

### Official Review · Reviewer_GESH · 2024-10-25

**Soundness:** 2
**Presentation:** 2
**Contribution:** 2
**Rating:** 3
**Confidence:** 5

**Summary:**

This paper proposes a framework called Tri-MoME for survival prediction. It extends an existing framework MoME using two notable modifications. The one is the integration of pathology reports into WSI and gene. The other is adopting an existing tool, Hyperbolic Representation Space (Desai et al., 2023), to capture the visual-semantic hierarchy between image and texts for learning better multi-modal representations. Experiments on four datasets demonstrate the effectiveness of the proposed Tri-MoME.

**Strengths:**

1. This work explores the integration of pathology reports for survival prediction. Survival prediction with triple domains (textual, gene, and image) remains under-explored in computational pathology.
2. The hyperbolic representation is leveraged in the framework to improve the multi-modal representation learning by capturing the visual-semantic hierarchy between images and texts.
3. The comparison to existing approaches and the ablation study are well-conducted to verify the performance of the proposed method.

**Weaknesses:**

Although this paper makes new investigations in computational pathology, such as hyperbolic representation and report integration, there are still substantial improvements to be made to get it published in well-known conferences like ICLR. My critical concerns are as follows:
- Writing: Some motivations are not elaborated clearly enough in the Introduction. For example, why did the authors choose to adopt MoME? From my understanding, CGM seems to be a better backbone for gene and WSI data, as shown in Table 1. Moreover, the explanation of the motivation behind using hyperbolic representation seems inefficient to readers, especially those unfamiliar with MERU (Desai et al., 2023). The authors are encouraged to improve it to make the paper self-contained and friendly to general readers.
- Experiments:
  - On page 2, line 99: The authors claim "our approach significantly enhances the ability of the model to capture the visual-semantic hierarchy". However, the paper fails to show the visual-semantic hierarchy the proposed model has captured. This is very important for readers to understand the mechanism and gain new insights,  as the visual-semantic hierarchy is one main thing that this paper wants to demonstrate.
  - To ensure the general use of the proposed Hyperbolic loss, the authors are encouraged to validate it using a different backbone beyond the MoME. By doing so, the readers would know the proposed Hyperbolic loss is a generally-better tool, rather than being only effective in MoME.
  - On page 2, line 72: The authors claim "xxx a single pass, enhancing efficiency and xxx". I am curious about the efficiency of the proposed Tri-MoMe, e.g., how are the model efficiency and the computational efficiency? is the framework more efficient than existing ones?

The authors are encouraged to resolve the critical concerns above to make this work sounder to the computational pathology community.

**Questions:**

My most questions are given in Weaknesses. The authors could respond to them.

My additional questions are as follows:
- Enabling the model to capture visual-semantic hierarchy using hyperbolic representation is first seen in MERU (Desai et al., 2023). The task that MERU focuses on is vision-language contrastive pretraining; and the visual-semantic hierarchy is well demonstrated. So, I am curious about whether the visual-semantic hierarchy can also be captured for a different task like supervised representation learning.
- Do the pathology reports contain survival labels? If not, how do the authors avoid it?
- The technical details of Expert Pool, especially its functionality, seem unclear to me. Could the author make them clearer? It is recommended to ensure the paper is self-contained.

I will be happy to raise my score if my concerns & questions can be addressed.

---

### Official Review · Reviewer_SskS · 2024-10-29

**Soundness:** 2
**Presentation:** 3
**Contribution:** 2
**Rating:** 3
**Confidence:** 4

**Summary:**

- **Motivation:** Survival prediction task is a complex task and it is unclear how to integrate Whole Slide Images (WSI), genomics, and text to predict survival. The integration of pathological reports can "guide" the model to learn regions relevant for survival prediction.
- **Main Technical contribution:** A hyperbolic loss framework (HyperSurv) is proposed to better "match" survival terms from pathology reports with relevant features from WSI. A mixture of experts for three modalities (Tri-MoMe) is proposed to handle WSI, genomics, and text reports.
- **Strengths:** One of the first works to use reports along with WSI to predict survival. Results are shown on 4 cancer types.
- **Weakness:** Flaws with evaluation framework and gain with adding text as a third modality is not obvious.

I recommended "reject" for this paper because while the method is novel, I am not fully confident about the gains with the method due to the limitations in the evaluation framework. Moreover, I am not able to see the benefit of adding text in addition to genomics and WSI for survival prediction as the alignment with text space is not explored.

**Strengths:**

**Clear presentation**
- The paper is generally clearly written and the details of the method are presented neatly.
- The figures are helpful in understanding the method

**Multiple datasets**
- The authors present results on multiple datasets to validate their model

**Various ways of testing survival**
- The authors validate their findings with KM curves as well as c-index.

**Weaknesses:**

**Concerns with evaluation frameworks**
- The authors use 5 fold-cross validation startegy without knowledge of tissue source sites to evaluate their model. This method has been shown to artificially inflate the performance of survival models (Kather) due to various demographic confounding factors ([1], [2]). Without the use of site-stratified splits, it is unclear how much performance can be attributed to confounding factors and actual signal. I request the authors to rerun their analysis with using site-stratified splits.
- The authors use CtransPath patch encoder. However, this model has been trained on TCGA and hence the same slides are used for pre-training and testing of the proposed method. 10+ foundation models for pathology have been proposed in the last year (Virchow 1/2 [3], GigaPath [4], UNI [5], CONCH [6], etc.), which are (1) significnatly better than CtransPath and are not trained on TCGA. I request the authors to (1) explain their rationale behind using CtransPath (2) try out one of the newer foundation models, which I believe should give a healthy improvement to the WSI only baselines.
- The authors optimize the hyperparameters of their model based on the validation set of their study (Table 3 for example shows that the best hyperparameters are chosen using the validation set). To avoid leaning towards test-set optimization of the model, I urge the authors to try out the WSI encoding part of their method on an independent test cohort, such as CPTAC-LUAD or CPTAC-UCEC.

**Marginal gains over baselines**
- The gain in performance with the proposed method seems small with respect to the standard deviations. I urge the authors to use statistical tests to support whether their method is significantly better than the baselines.

**Missing baselines**
- SurvPath (CVPR'24) [8] is an important baseline that is missing. Just like the authors' method, SurvPath uses gene-groupings. The authors know of this baselines as they cite them in lines 140-141.
- While it is commendable that the authors present KM curves to show patient stratification with their model, I would like to see KM curve comparisons with the next best baselines (for example CGM).

---
**References**
- [1]: Howard, Frederick M., et al. "The impact of site-specific digital histology signatures on deep learning model accuracy and bias." Nature communications 12.1 (2021): 4423.
APA
- [2]: Vaidya, Anurag, et al. "Demographic bias in misdiagnosis by computational pathology models." Nature Medicine 30.4 (2024): 1174-1190.
- [3]: Vorontsov, Eugene, et al. "A foundation model for clinical-grade computational pathology and rare cancers detection." Nature medicine (2024): 1-12.
- [4]: Xu, Hanwen, et al. "A whole-slide foundation model for digital pathology from real-world data." Nature (2024): 1-8.
- [5]: Chen, Richard J., et al. "Towards a general-purpose foundation model for computational pathology." Nature Medicine 30.3 (2024): 850-862.
- [6]: Lu, Ming Y., et al. "A visual-language foundation model for computational pathology." Nature Medicine 30.3 (2024): 863-874.
- [7]: Xu, Yingxue, and Hao Chen. "Multimodal optimal transport-based co-attention transformer with global structure consistency for survival prediction." Proceedings of the IEEE/CVF International Conference on Computer Vision. 2023.
APA
- [8]: Jaume, Guillaume, et al. "Modeling dense multimodal interactions between biological pathways and histology for survival prediction." Proceedings of the IEEE/CVF Conference on Computer Vision and Pattern Recognition. 2024.
APA

**Questions:**

- Typo in line 98.
- Typo in ine 115.
- Authors seem to convey that UNI, CONCH and ViT are different architectures, but this is not correct. UNI and CONCH are different ViT architectures.
- Lines 125-129 needs references for use of genomics for survival prediction and associated architectures ([Chen et al] for SNN)

---

### Official Review · Reviewer_hByA · 2024-11-06

**Soundness:** 3
**Presentation:** 3
**Contribution:** 3
**Rating:** 5
**Confidence:** 5

**Summary:**

This paper proposes a new survival prediction model, HyperSurv, integrating Whole Slide Images (WSIs), genomic data, and pathology reports within a multimodal framework. Utilizing hyperbolic geometry, specifically the Lorentz manifold, the model enforces two constraints—the Feature-pooled Entailment Constraint (FEC) and Text Genericity Constraint (TGC)—to capture the hierarchical relationship between WSIs and pathology reports. Extensive experiments across TCGA datasets show that HyperSurv achieves superior performance over existing methods, underscoring the added predictive value of pathology reports and the effectiveness of hyperbolic embeddings for survival prediction.

**Strengths:**

- **Well-Defined and Relevant Problem:** The paper addresses an important and timely problem, focusing on modeling the relationship between pathology reports and images, which has significant implications for advancing diagnostic accuracy.

- **Effective Integration of Multiple Modalities:** The integration of three modalities—textual pathology reports, medical images, and structured data—is well-motivated and demonstrates the authors' commitment to capturing complex, multimodal relationships, which is highly valuable for clinical applications.

- **Clear and Well-Structured Presentation:** The paper is well-written and logically organized, making the proposed method and findings easy to understand. The clear structure allows readers to follow the motivation, methodology, and conclusions seamlessly.

- **Comprehensive Experiments:** The authors present extensive experimental results, which convincingly demonstrate the performance advantages of the proposed approach over existing methods, highlighting its effectiveness and robustness.

- **Innovative Use of Hyperbolic Space:** The application of hyperbolic space to model relationships between pathology reports and images is intriguing, offering a unique perspective that aligns well with the hierarchical and relational nature of clinical data. This approach adds an interesting dimension to the study and may inspire further research in this direction.

**Weaknesses:**

- **Limited Empirical Support for Pathology Report-Image Relationship Claims:**
Although the paper claims to effectively model the relationship between pathology report terms and corresponding images—specifically addressing that a single pathology term may map to multiple images and that pathology terms convey more generic concepts—this contribution is only discussed in the methods section. There are no experimental or visual results to substantiate this claim, leaving it unvalidated in the results section. Providing empirical or visualization evidence would strengthen this aspect of the paper.

- **Incomplete Multimodal Relationship Modeling:**
While the paper addresses the relationship between text and pathology images, it does not appear to effectively capture the relationships between text and gene data or between images and gene data. Incorporating these additional interactions could enhance the model's ability to represent the complex dependencies across all three modalities and improve its overall performance in clinical settings.

- **Limited Implementation Details:**
The paper lacks sufficient implementation information. It does not provide the experimental code, nor does it include details or examples on the preprocessing steps for pathology reports. This lack of transparency makes it challenging for others to replicate the study and validate the results, limiting the paper's practical impact and reproducibility.


In summary, although the problem and approach described by the authors are meaningful, additional technical details and experimental results are needed to fully demonstrate the effectiveness of their method. Providing more comprehensive implementation information and empirical evidence would strengthen the paper’s contributions and validate its claims.

**Questions:**

- **Provide Empirical or Visual Evidence for Pathology Report-Image Relationship:**
To validate the claimed relationship between pathology report terms and corresponding images, the authors could include experimental or visual results demonstrating this mapping. For example, visualizations of how specific pathology terms correspond to different image patches or the cross-attention maps would provide valuable insights and empirical support for this contribution.

- **Enhance Multimodal Relationship Modeling:**
To improve the completeness of the multimodal approach, the authors might consider modeling the relationships not only between text and images but also between text and gene data and between images and gene data. Since different genes might also relate to different pathology terms, do the authors consider utilizing this relationship? Incorporating these additional interactions could strengthen the model's representation across modalities, making it more effective in clinical applications.

- **Increase Implementation Transparency:**
Including the experimental code and detailed preprocessing steps for pathology reports would make the study more reproducible and accessible for other researchers. Sample instances or detailed guidelines for data preparation of the pathologist report and gene expression would also assist in replicating the results, ultimately enhancing the paper’s practical impact and credibility.

- **Expand on Technical Details and Experimental Validation:**
To strengthen the paper’s claims, the authors could provide additional technical insights and experimental evidence, especially regarding the model’s effectiveness in capturing multimodal relationships. This would reinforce the significance of their contributions and better demonstrate the approach’s value in practical settings.

---

### Official Review · Reviewer_ffFM · 2024-11-08

**Soundness:** 1
**Presentation:** 2
**Contribution:** 1
**Rating:** 3
**Confidence:** 5

**Summary:**

This work presents Tri-MoME for trimodal fusion of pathology, genomics, and text reports for survival analysis in TCGA. Tri-MoME is based on the multimodal mixture of experts architecture (MoME) extending from Xiong et al. 2024 with additional hyperbolic constraints. Evaluation is done on 4 TCGA cancer cohorts.

**Strengths:**

- Interesting idea of extending multimodal survival analysis with hyperbolic constraints.
- Good visualizations and figures in communicating the Tri-MoME framework.

**Weaknesses:**

**Weaknesses**
- This work misses comparison with current early fusion methods published in the last year: SurvPath in CVPR'24, PIBD in ICLR'24, and MMP in ICML'24. In addition, SurvPath, PIBD, and MMP propose new techniques for reducing complexity of cross-attention modeling in multimodal pathology-genomics data, which can be easily extended to text.
- How well do other methods perform with also text integration? There is missing investigation if method is performing better because it is including three modalities. All other comparisons have only one or two.
- How well does unimodal text and bimodal fusion (pathology+genomics, genomics+text, pathology+text) perform? There is missing investigation on the need to train with all three modalities within MoME. The bimodal fusion baselines should also be based on the MoME architecture with hyperbolic constraints. The pathology+genomics baseline is especially needed for apples-to-apples comparison with other comparisons that only use two modalities.
- How much does having a MoE-based architecture matter? There is missing investigation on comparing MoE design of Tri-MoMe with conventional Transformer architecture that can integrate tokens across long contexts in a straightforward manner. The problem in modeling long context may not matter as much with recent works like PIBD and MMP reducing WSIs into a bag of prototypes, which Transformers can easily perform cross-attention on.
- Not clear what prognostic information is captured from TCGA pathology reports using Tri-MoME. TCGA pathology reports contain very little information on observed morphological features. Features in TCGA pathology reports that would typically be prognostic would be patient demographic (age, sex), tumor size, tumor grade, and tumor stage - these features are more easily modeled as tabular attribute data that can be introduced into modeling via late fusion. A fair comparison would assess Tri-MoME and other models with pathology+genomics+clinical metadata (age, sex, grade, stage) fusion. The method feels too complex and there is missing investigation and analysis on what novel multimodal interactions are being learned by introducing a complex architecture.
- **We should stop using CTransPath anymore in TCGA survival analysis due to data contamination.** CTransPath is trained on TCGA. Using PLIP or CONCH as encoders make much more sense, especially with text integration. It is becoming increasingly more unethical to use pretrained models in such a manner with our growing awareness of data contamination.

**Suggestions**

I think the rating of this paper can be greatly improved if additional experimentation was performed in Tri-MoME for apples-to-apples comparison with other baselines (current SOTA with also text, Tri-MoME with pathology+genomics, Tri-MoME with clinical attributes, unimodal text survival analysis). Using CTransPath is problematic, but evaluation with a different encoder (preferably a vision-language encoder like PLIP/CONCH based on the theme of this work) would also greatly improve the soundness of this paper. Text integration is an interesting idea for improving multimodal cancer prognosis, but I don't think using TCGA pathology reports introduce any unique information that would also not be captured in clinical metadata. I think this aspect can be improved if the authors can show meaningful insights into pathology-genomics-text interactions being learned for survival analysis.

**Questions:**

How were the splits in Tri-MoME created?

**Details Of Ethics Concerns:**

This work uses a pretrained encoder that is trained on the test dataset. It is becoming increasingly more unethical to use pretrained models in such a manner with our growing awareness of data contamination.

---

### Note · Authors · 2024-11-14

I have read and agree with the venue's withdrawal policy on behalf of myself and my co-authors.